# Biofilm-Based Biocatalysis for Galactooligosaccharides Production by the Surface Display of β-Galactosidase in *Pichia pastoris*

**DOI:** 10.3390/ijms24076507

**Published:** 2023-03-30

**Authors:** Tianpeng Chen, Shimeng Wang, Huanqing Niu, Guanjia Yang, Sinan Wang, Yuqi Wang, Chaowei Zhou, Bin Yu, Pengpeng Yang, Wenjun Sun, Dong Liu, Hanjie Ying, Yong Chen

**Affiliations:** 1National Engineering Research Center for Biotechnology, College of Biotechnology and Pharmaceutical Engineering, Nanjing Tech University, Nanjing 211816, China; chentianpeng@njtech.edu.cn (T.C.);; 2State Key Laboratory of Materials-Oriented Chemical Engineering, College of Biotechnology and Pharmaceutical Engineering, Nanjing Tech University, Nanjing 211816, China

**Keywords:** yeast surface display, *Pichia pastoris*, biofilm, galactooligosaccharides, β-galactosidase

## Abstract

Galactooligosaccharides (GOS) are one of the most important functional oligosaccharide prebiotics. The surface display of enzymes was considered one of the most excellent strategies to obtain these products. However, a rough industrial environment would affect the biocatalytic process. The catalytic process could be efficiently improved using biofilm-based fermentation with high resistance and activity. Therefore, the combination of the surface display of β-galactosidase and biofilm formation in *Pichia pastoris* was constructed. The results showed that the catalytic conversion rate of GOS was up to 50.3% with the maximum enzyme activity of 5125 U/g by screening the anchorin, and the number of the continuous catalysis batches was up to 23 times. Thus, surface display based on biofilm-immobilized fermentation integrated catalysis and growth was a co-culture system, such that a dynamic equilibrium in the consolidated integrative process was achieved. This study provides the basis for developing biofilm-based surface display methods in *P. pastoris* during biochemical production processes.

## 1. Introduction

Galactooligosaccharides (GOS) are functional oligosaccharides with a molecular structure consisting of one to seven galactose groups attached to a galactose or glucose molecule, i.e., Gal-(Gal) n-Glc/Gal (n = 0–6) [1]. GOS are an excellent source of nutrients and could be an effective proliferation factor for beneficial bacteria such as *Bifidobacterium* and *Lactobacillus acidophilus* in the human intestinal tract, which could improve the functions of digestion and absorption of the intestinal tract of infants and children. Moreover, there are trace amounts of GOS found in animal milk and human breast milk [2]. Hence, GOS were approved to be added to infant formula as important nutrients. Lactose was generally used as the substrate for the industrial production of GOS, which was catalyzed by the trans-glycosylation of β-galactosidase [3]. In nature, microorganisms such as *Aspergillus oryzae* [4], *Bacillus circulans* [5], *Bifidobacterium infantis* [6], and *Kluyveromyces lactis* [7] are reported to have β-galactosidase glycosyltransferase activity [8,9]. In particular, the immobilized β-galactosidase from *A. oryzae* is relatively inexpensive and readily available, with the high specificity and trans-glycosylation activity, as well as being reusable and continuous, compared to the free enzyme [4,10]. However, the deficiencies such as the loss of enzyme activity and the isolation and purification of products have limited the progress of the traditional immobilized enzyme catalysis of β-galactosidase.

Surface display has gained worldwide attention as an available strategy for circumvention of the above deficiency during its green and efficient process [11]. The surface display of enzymes could save the steps of enzyme extraction and purification and the overall cost. For instance, the surface display of α-galactosidase derived from *Aspergillus niger* was carried out in *Pichia pastoris* for isomaltooligosaccharide (IMO) production from 30% maltose feedstock with the conversion rate of approximately 49% and three repeated catalysis cycles [12]. Hence, we used the surface display technique of *P. pastoris* in this study because it has well-established protein folding and secretory expression mechanisms. It can perform many of the post-translational modifications found in eukaryotes and is a good host choice due to its ease of handling, rapid reproduction, and high expression of heterologous proteins.

However, environmental stress on the cells and cell autolysis in the fermentation broth could result in reducing enzyme activity and catalytic efficiency in surface-displayed strains. To address this issue, biofilm-based immobilized fermentation has been proposed as an alternative to free-cell fermentation owing to advantages such as the protection provided by the multicellular aggregate matrix, enhanced cell viability, and repeated use of the cells [13]. Moreover, these cell cultures are adapted to adverse external environments such as high osmolarity, heat shock, oxidative stress, and nutrient deficiencies [14]. Based on this, some microorganisms such as *Clostridium acetobutylicum*, *Escherichia coli,* and *Saccharomyces cerevisiae* were used for immobilized batch or continuous (repeated-batch) fermentation effectively [15,16,17,18,19,20]. More specifically, we aimed to understand the effects of cell wall proteins, adhesion factors, and signaling molecules on biofilms [21,22]. For example, we previously identified the biofilm-related gene *PAS_chr1-3_0226* (Gene ID: 8196458) in *P. pastoris*, which encodes a GPI protein with high homology to GAS1p in *S. cerevisiae* [23], that was anchored to the cell surface and could counteract intracellular pressure and maintain normal cell morphology [24]. After the knockout of *PAS_chr1-3_0226*, the results showed increased levels of chitin and the greater resistance to lysozyme, suggesting the gene was associated with cell adhesion [25]. Gene knockout was shown to enhance the efficiency of surface display in *P. pastoris*; therefore, we selected this gene for studies related to biofilm and surface display.

In this study, we aimed to solve the potential deficiency of biofilm-immobilized fermentation and surface display methods by addressing the problem of reduced enzyme activity and catalytic efficiency together. To achieve this, genetic engineering was performed on *P. pastoris* to knock out the *PAS_chr1-3_0226* gene and enhance biofilm formation (Figure 1). Then, biofilm-based immobilized fermentation was constructed by using cotton fiber material as the carrier. In addition, the screenings of different anchorins resulted in the increased efficiency of the β-galactosidase surface-displayed strain. The combination of the surface-displayed system and biofilm-immobilized fermentation enabled continuous (repeated-batch) catalysis, which significantly shortened the catalytic cycle and improved production efficiency. Here, we successfully demonstrated for the first time the efficient biochemical production of GOS using biofilm-immobilized surface-display enzyme catalysis in industrial conditions.

## 2. Results and Discussion

### 2.1. The Impact of PAS_chr1-3_0226 Gene Deletion on Biofilm Formation in P. pastoris

In this study, the gene *PAS_chr1-3_0226* was deleted from the *P. pastoris* GS115 genome by homologous recombination. The knockdown effect on the biofilm was observed by significantly darker crystalline violet staining in the Δ0226 strain when compared to that of the WT strain (Figure 2a). Readings were taken at OD_570_, where the biofilm of Δ0226 increased by 56% (Figure 2b), while the YPD medium was slightly better than the fermentation medium for biofilm production. We also observed that when the Δ0226 strain was grown on YPD plates, there was little effect on its growth (Figure 2c).

The cell infestation ability assay was used to observe the ability of the recombinant strains to adhere to the YPD plates. The WT strains had a flat and smooth surface after rinsing and left little residue on the plates. Whereas, the recombinant strains all had a thick wall around the edges of the cells, with the GS115-Δ0226 strain showing a significant retention effect on the plates. The infection range of strain Δ0226 on the plate was larger than that of the WT strain, indicating that gene knockout improved the ability of infection (Figure 2d). This result was consistent with the tendency of biofilm formation in 96-well plates, suggesting that GS115-Δ0226 has a facilitative effect on the growth of biofilms.

### 2.2. Biofilm Formation on Carriers and Cell Morphological Changes

To further explore the amount of biofilm formation, the carriers were removed after fermentation of 24 h, the surface medium and free cells were removed with PBS, and the biofilms were fixed with 2.5% glutaraldehyde and immediately freeze-dried before observation with SEM. A few cells could be seen on the carrier of the WT strain but they did not form clusters (Figure 3a(II,V)), while the GS115-Δ0226 strain had significantly enhanced biofilm formation and the adsorption on the vector increased, with clusters of cells forming the biofilm (Figure 3a(III,VI)). These results further demonstrated that the Δ0226 strain had a facilitative effect on biofilm formation. The enhanced ability of biofilm formation laid the foundation for the strategy of biofilm-based surface display.

In order to investigate the microstructure changes of cell morphology, a TEM test was carried out. Generally speaking, the beta-1,3-glucanosyltransferase encoded by *PAS_chr1-3_0226* was required for cell wall assembly in *Pichia pastoris* [24,26]. The results of TEM showed that the morphology of WT was normal with a smooth surface (Figure 3b(I,III)). Meanwhile, the substances similar to biofilm matrix were observed on the surface of strain Δ0226 (Figure 3b(II,IV)). These results were consistent with the conclusion that the deletion of *PAS_chr1-3_0226* could enhance biofilm formation in Figure 2. The surface of strain Δ0226 was rough and the specific surface area was increased, which could be beneficial to form more anchorin on the surface of the cell wall in the following surface display.

### 2.3. Construction of Strains for Surface Display

The β-galactosidase surface display system was constructed from three different anchorins with the *P. pastoris* methanol-inducible plasmid pPIC9K, which contained an α-signal peptide (Figure 1). The active center of β-galactosidase is in the C-terminus and therefore, β-galactosidase was fused to the C-termini of the anchorins Pir1p, Aga2p, and Flo1p. In order to prevent the signal peptide of the protein from being excised by the signal peptidase during cellular secretion, the signal peptide of both was removed. The terminator of the anchorins was also removed and a FLAG tag was added in front of the terminator of *lacA* for subsequent immunofluorescence and flow cytometry detection. All three recombinant strains were constructed successfully.

### 2.4. Expression of Surface-Displayed β-Galactosidase

The β-galactosidase in situ staining kit was used to verify the successful expression of β-galactosidase on the cell surface using X-Gal as a substrate, in which a dark blue product was observed as shown in Figure 4a. We could see the darkest blue color in sample Ⅲ, indicating that strain Δ0226-Pir1p-lacA had the higher expression of β-galactosidase.

### 2.5. Enzyme Activity of Surface-Displayed β-Galactosidase

β-Galactosidase is an extracellular enzyme that is secreted out of the cell to the supernatant. Whereas, to achieve the surface display of β-galactosidase, it should stay on the cell surface. The effect of different substrates on enzyme activity was first screened by measuring the enzyme activity of the precipitate. The highest enzyme activity was measured when 800 μL *o*NPG was added to the sample, reaching 1828 U/g enzymatic activity (Figure 4b). Therefore, 800 μL *o*NPG was selected for the subsequent enzyme activity assays to determine whether the cell surface and supernatant had enzymatic activity. Subsequently, the samples of 1 mL were taken every 24 h to react in a water bath at pH 6.0 and 50 °C. The enzyme activity was determined in the precipitate and supernatant of the surface-displayed strains compared with that of the free type strain. The enzyme activity of the precipitate was defined by U/g, whereas that of the supernatant was defined as U/mL. We found that the Δ0226-Pir1p-lacA precipitate had the highest enzyme activity (5125 U/g) of the strains measured; however, Δ0226-Flo1p-lacA and Δ0226-Aga2p-lacA had relatively weak enzyme activities of 3389 U/g and 2643 U/g, respectively (Figure 4c). We also found that the enzymatic activity of Δ0226-Pir1p-lacA, Δ0226-Flo1p-lacA, and Δ0226-Aga2p-lacA was higher than that of WT-Pir1p-lacA, WT-Flo1p-lacA, and WT-Aga2p-lacA, indicating that the surface display efficiency was increased in the Δ0226 knockout strain, which allowed for more anchorin expression on the cell wall of *P. pastoris* (Appendix A). This result was consistent with those of the in situ staining experiments, so we selected the Δ0226-Pir1p-lacA strain with the highest enzyme activity for subsequent experiments.

We also performed free expression of β-galactosidase as a supernatant enzyme activity control. When we measured the supernatant enzyme activity of the recombinant surface-displayed strains and the free type strain, we found that the free expressed enzyme activity reached a maximum of 206 U/mL at 96 h, while the highest enzyme activity of all recombinant surface-displayed strains was 18 U/mL at 144 h (Figure 4d), which was much lower than the free expression. This result suggested that most of the surface-displayed enzyme was on the cell surface and only a minimal amount was present in the supernatant, probably due to the natural lysis of the cells and the release of some of the enzymes into the fermentation broth.

### 2.6. Verification of Surface-Displayed β-Galactosidase by Immunofluorescence Microscopy and Flow Cytometry

A 24 bp FLAG tag (DYKDDDDK) was added in front of the target protein β-galactosidase terminator to accurately localize the fusion protein within the cell in immunofluorescence (IF) experiments. The results of immunofluorescence microscopy further confirmed that the fusion protein localized to the surface of *P. pastoris*. Fluorescence microscopy showed that the cell surface emitted a visible signal (Figure 5a(Ⅳ)), whereas the no fluorescence was observed in the control (Figure 5a(Ⅰ)), indicating that the flag tag on β-galactosidase successfully bound to the antibody and glowed under FITC excitation light successfully. Flow cytometry showed that a significant rightward shift in the strain Δ0226-Pir1p-lacA in the peak plot of the cytometry histogram compared to the control. Meanwhile, an increase in fluorescence intensity and a high display efficiency of 80.63% were detected. This experiment had demonstrated that the fusion proteins with high fusion efficiency were covered on the surface of *P. pastoris* (Figure 5b).

### 2.7. Characterization of Surface-Displayed β-Galactosidase Activity

We next tested the effect of different temperatures, pH, and metal ions on the relative activity and thermal stability of surface-displayed β-galactosidase. The surface-displayed β-galactosidase activity was highest at 50 °C and reached 94.2% and 75.9% at 60 °C and 70 °C (Figure 6a), respectively. This was similar to that of *A. oryzae* and *B. coagulans* reported by Gerhartz et al. [27,28], but 10 °C higher than the optimum reaction temperature of the immobilized enzyme previously reported by Matella et al. [29]. As the temperature continued to increase, the relative activity of surface-displayed β-galactosidase decreased, indicating a loss of enzyme activity. In particular, the thermostability of the surface-displayed β-galactosidase was important in the catalytic process. The enzyme was most stable at 50 °C and 40 °C; after 120 min at 50 °C, it maintained 84.6% of the enzyme activity. However, at temperatures above 60 °C, the enzyme activity dropped abruptly and was almost inactive (Figure 6d).

The maximum enzymatic activity of β-galactosidase was demonstrated at pH 5.0 and was maintained at 93.5% and 68.6% at pH 6.0 and pH 7.0, respectively. The enzyme activity dropped sharply when was below pH 5.0 and above pH 7.0, such that the maximum activity was only 19.8% (Figure 6b). Although the optimal enzyme activity was achieved at pH 5.0, it was decreased to 36.5% after 120 min. As shown in Figure 6e, pH 6.0 was the most stable. Hence, it was chosen as the catalytic pH. The optimum pH for β-galactosidase was similar to that of the genetically recombinant strains studied by Katrolia et al. [30] and Lite et al. [31]. This indicated that either too high or low a pH had a significant effect on the enzyme displayed on the surface. We found that the half-life of the surface-display enzyme was still above 70% enzyme activity at 24 h, indicating its stability and role in later catalysis events (Figure 6f).

Metal ions can change the ionic strength of solutions, which in turn affects the ionization of amino acids and can change the protein structure. In addition, metal ions can affect the active site of enzymes, thereby affecting the enzymatic reaction. We found that different concentrations of metal ions had little effect on overall enzyme activity, where Ca^2+^, Mn^2+^, and Na^+^ at 5, 25, and 50 mmol/L all promoted enzyme activity (Figure 6c). Indeed, low concentrations of Mg^2+^ promoted enzyme activity, while high concentrations of Mg^2+^ slightly inhibited enzyme activity. Zn^2+^, Co^2+^, Fe^3+^, Cu^2+^, and Fe^2+^ had inhibitory effects on enzyme activity, with Zn^2+^, Fe^3+^, and Cu^2+^ having the most significant inhibitory effect on enzyme activity. These results indicated that some metal ions such as Ca^2+^, Mn^2+^, and Na^+^ can act as cofactors of β-galactosidase to enhance its activity, while others such as Cu^2+^ and Fe^3+^ can induce structural changes in the enzyme that inhibit its activity.

### 2.8. Optimizing the Process of Increasing GOS Yields

To ensure that the free and immobilized surface-display cells had the same amount of enzyme, both dry cells need to be weighed. The dry weight of the free cells was 0.92 g/100 mL and the immobilized cells was 1.72 g/100 mL, suggesting the amount of enzyme added for catalysis was 1.87-fold higher than that of the immobilized cells (Figure 7a). Cells in biofilm-based fermentation could produce more products during the same incubation time. Moreover, the biofilm-immobilized fermentation process maintained the cell activity, slowed down cell degradation, and improved catalytic efficiency.

β-galactosidase could convert the high concentration of lactose substrate into GOS. Therefore, the initial lactose concentration is an important factor that affects the production of GOS. The reaction was carried out at 50 ℃ at lactose concentrations of 300–600 g/L, which showed that at higher concentrations of lactose, GOS production increased continuously with the highest yield achieved at a lactose concentration of 500 g/L. This suggested that the glycosyltransferase activity increased with increasing concentrations of lactose substrate. As the lactose concentration rose, the conversion rate slowly decreased. The lactose was not completely soluble, which could have affected the final trans-glycosylation reaction. The yield of the product was measured by sampling the liquid phase every 3 h, and the yield increased steadily during the first 12 h before it reached a maximum of 251 g/L at 12 h (Figure 7b), with a conversion rate of 50.3%. No significant change in yield was observed with an increasing reaction time, so 500 g/L for 12 h was used as the optimal experimental condition.

Different volumes of BMMY medium were added to the catalytic reaction to observe which condition ensured the growth of cells without affecting the catalytic activity. Lactose was added in citrate buffer at 20%, 50%, 80%, and 100% of BMMY medium and re-added every 12 h to observe the effect of GOS yield in four consecutive batches. The GOS yield was higher with citrate buffer and 20% BMMY, where the yield was almost the same as that collected from the first two batches of catalysis, while the third batch the GOS yield was slightly higher in a system with 20% BMMY medium than in the citrate buffer (Figure 7c). This result showed that adding 20% BMMY medium ensured that cells remained enzymatically active during growth. The catalytic efficiency was inhibited when more than 20% of the volume of BMMY was added, presumably due to competition between growth and enzyme catalysis. Therefore, a buffer that was 20% of the volume of BMMY medium was used as the buffer.

### 2.9. Free and Immobilized Cell Biocatalytic GOS Production

Immobilized cotton fiber material was selected as the carrier [21]. Immobilized cells and free cells catalyzed 500 g/L of lactose at 40 °C. The product yield was measured using HPLC. After each batch of reaction, the buffer was removed and fresh culture with lactose was added. The number of reusable batches was recorded.

High concentrations of lactose led to trans-glycosylation reactions in which glucose, galactose, transferred disaccharides, trisaccharides, and tetrasaccharides were produced. Table 1 shows that in the first batch of catalysis, there was almost no difference between the yields of free and immobilized cells, with immobilization slightly higher than that of free cells. The highest proportion of GOS trisaccharides, which accounted for 23.6% of the composition, was the same as found by Gao Xin et al. [32]. Moreover, when the free cells were reused 8 times, the GOS yield dropped to 75%; the first 4 batches maintained a yield above 90%, whereas, the 5th batch showed reduced enzyme activity and the yield dropped more rapidly (Figure 7d). The immobilized cells were reused 10 times, which was 2 times more than the free cells, demonstrating that the addition of 20% BMMY could maintain cell activity and slow down enzyme activity loss. When the yield was dropped to 71%, the carriers were taken out and the fresh medium was added to reaction for 48 h to restore cell viability. A second catalysis was then conducted, which was stopped when the yield dropped to 73% at the end of the 7th batch and the recovery was continued for 48 h. A total of 6 catalysis repeats was achieved at 3rd batch, of which the recovery was stopped once the yield was dropped to 63%. The immobilized cells maintained catalytic activity for a total of 23 times in 3 batches, which is 15 times more than free cells (Figure 7e). The catalytic time and efficiency were both important aspects in the industrial production process. A total of 120 h of growth time was needed for every batch in free fermentation with 8 catalytic repeats. Although 120 h of growth time was still needed in the initial immobilized fermentation, the next growth time was instead a time of 48 h with the total of 23 catalytic times in 3 batches. The productivity of immobilized and free fermentation were 0.44 g/L/h and 0.33 g/L/h, respectively, showing how the production efficiency was significantly improved. Furthermore, the dry weight of immobilized fermentation cells was nearly twice than that of the free type, which is a great advantage in industrial production that was achieved. Overall, biofilm-based biocatalysis using surface display obtained a higher efficient productivity of GOS, shortened culture time, and the persistent cell activity.

## 3. Materials and Methods

### 3.1. Strains and Plasmids

*P. pastoris* GS115 was selected for this study. The gene *lacA* (GenBank: FM955406.1) coded β-galactosidase, which was synthesized by Tsingke Biotechnology Co., Ltd. (Beijing, China). The anchorins Pir1p, Aga2p, and Flo1p were all derived from *S. cerevisiae* S288c [33,34,35]. The plasmids were derived from the *P. pastoris* inducible plasmid pPIC9K and the expression plasmid pPICZαA. pPIC9K was used for the construction of surface-display strains and pPICZαA was used for the knockdown of the gene *PAS_chr1-3_0226* (Table 2).

### 3.2. Media and Growth Conditions

The host for the molecular manipulation in this study was *E. coli* DH5α, which was grown in LB medium composed of yeast extract (5 g/L), peptone (10 g/L), and NaCl (5 g/L). *P. pastoris* was cultured in Yeast Extract Peptone Dextrose (YPD) medium composed of yeast extract (10 g/L), peptone (20 g/L), and glucose (20 g/L). Solid media were prepared in all cases by adding 1.5% (*w*/*v*) agar. Buffered Minimal Glycerol YP (BMGY) medium was composed of yeast extract (10 g/L), peptone (20 g/L), biotin (0.0004 g/L), 10% glycerol, K_2_HPO_4_·3H_2_O (3 g/L), KH_2_PO_4_ (11.8 g/L), and Yeast Nitrogen Base (13.4 g/L) (YNB). Buffered Minimal Methanol Medium (BMMY) was composed of yeast extract (10 g/L), peptone (20 g/L), biotin (0.0004 g/L), K_2_HPO_4_·3H_2_O (3 g/L), KH_2_PO_4_ (11.8 g/L), YNB (13.4 g/L), and methanol (AR, 5 mL/L. The initial pH was adjusted to 6.5 ± 0.2.

### 3.3. Construction of Gene Knock-Out and Surface-Displayed Strains

The homologous recombination was selected for the construction of a *P. pastoris* GS115 strain mutant as previously described [36]. The fragment of bleomycin was amplified by PCR using BleoR-F/R primers with the pPICZαA plasmid as a template. Similarly, the upper and lower homology arms (500 bp) of *PAS_chr1-3_0226* were obtained using primers from the genome of *P. pastoris* GS115. The sequences of primers are listed in Table 3. The *PAS_chr1-3_0226* gene in the GS115 genome was replaced by the gene of bleomycin successfully.

The encoding genes of anchorin Pir1p, Aga2p, and Flo1p were amplified by PCR from *S. cerevisiae* S288c with the fusion of the *lacA* gene, which was inserted into the pPIC9K plasmid. Competent cells of GS115-Δ0226 were prepared using the sorbitol method, and then the recombinant plasmids were electroporated into GS115-Δ0226 strain.

### 3.4. Characterization of Biofilm Formation

#### 3.4.1. Biofilm Formation on Plastics

A crystal violet (CV) assay was performed to measure biofilm formation with minor modifications [37]. Yeast strains were grown in YPD and fermentation liquid media overnight at 30 °C with 150 rpm. After collection, washing, and discarding supernatant, the cells were resuspended in YPD and fermentation medium at an OD_600_ = 1. Cell suspensions (20 μL) were transferred to a 96-well microtiter plate (Corning, Kennebunk, ME, USA) containing 180 μL medium per well. The plate was then incubated for 72 h at 30 °C. Four replicate wells were used for each treatment. The biofilm-containing wells were washed twice with 200 μL PBS to remove free cells, after which the biofilms were stained with 200 μL 0.1% CV solution for 15 min at room temperature (about 25 °C), followed by repeated washing of the wells with PBS. Then, 200 μL 33% acetic acid was added to each well and the plate was incubated at room temperature for 20 min with slight shaking to elute the CV. Finally, the absorbance was measured at 570 nm using a microplate reader (SpectraMax^®^ iD5; Molecular Devices, San Jose, CA, USA).

#### 3.4.2. Cell Growth and Infiltration Capacity Analysis

*P. pastoris* strains were grown in YPD at 30 °C for 2 days. After collection, washing and discarding supernatant, the cells were resuspended in YPD at an OD_600_ = 1 and used to make 10^−1^, 10^−2^, and 10^−3^ gradient dilutions in sterile water. Each dilution (1 μL) was then dropped on YPD agar and incubated for 24 h at 30 °C.

The GS115 and GS115-Δ0226 strains were isolated in 5 mL/50 mL centrifuge tubes and cultured overnight at 30 °C and 250 rpm. The optical density of the strains was measured with a UV spectrophotometer and diluted to OD_600_ = 1 in YPD. A total of 10 μL was cultured in a YPD plate at 30 ℃ for 3–4 days. The cultured strains were slowly washed with running water until the cells could not be eluted and observed to remain on the medium [38].

#### 3.4.3. Characterization of Scanning Electron Microscopy and Transmission Electron Microscopy

The GS115 and GS115-Δ0226 strains were incubated in BMMY fermentation medium on cotton fiber material for 24 h. After immobilization, the carriers were removed and cut into suitable-sized squares and repeatedly rinsed 2–3 times with PBS buffer to ensure that no medium or yeast remained on the surface of the carriers. Immediately afterward, the samples were fixed in 2.5% glutaraldehyde with triton for 30 min at room temperature, and then washed 2–3 times with PBS and placed in a −80 °C freezer before they were dried overnight in a freeze drier (Labconco, Fort Scott, Kansas, USA). Freeze-dried samples were sputter-coated with gold and observed with a scanning electron microscope (TM3000, Hitachi, Tokyo, Japan) and transmission electron microscope (HT7820, Hitachi, Tokyo, Japan) [13].

### 3.5. Surface-Displayed β-Galactosidase Assay

#### 3.5.1. β-Galactosidase In Situ Staining Assay

An in situ β-galactosidase staining kit is a tool for the in situ staining of cells for β-galactosidase detection. The mutant strains were induced with methanol in BMMY medium for 48 h. Then, the fermentation broth was centrifuged at 6000 rpm for 5 min to remove the supernatant. The cells were washed twice with PBS and centrifuged again to remove the supernatant. An aliquot of 500 μL staining solution (Beyotime Biotechnology Co., Ltd., Shanghai, China) was added to each 1.5 mL centrifuge tube. The cells were incubated at 50 °C until the color stabilized and the cell morphology was observed under an optical microscope.

#### 3.5.2. β-Galactosidase Enzyme Activity

The enzymatic activity of β-galactosidase was determined as described previously [32]. The *P. pastoris* GS115 strain and the positive transformants were incubated overnight in 5 mL YPD, which was then used to inoculate 25 mL BMGY in conical flasks. These cultures were incubated at 30 °C for 21–24 h until OD_600_ = 2–6. The supernatants were removed by centrifugation at 6000 rpm for 5 min, and the precipitate was washed twice with sterile water before it was transferred to 100 mL BMMY. The enzyme activity was measured every 24 h by taking 1 mL of the solution after the expression was induced with 1% methanol.

The enzyme activity was measured by hydrolysis of o-nitrophenyl-β-galactoside (*o*NPG). The different volume of *o*NPG liquid was added to the precipitate of 1 mL to determine enzyme activity. The extract was centrifuged to remove the supernatant, and the cells were washed twice with PBS to remove any residual supernatant. A total of 1.2 mL citrate buffer at pH 6.0 and 800 μL 20 mmol/L *o*NPG were added to the bacterial precipitate and incubated in a water bath at 50 °C for 10 min. After the reaction, 1 mL of 1 mol/L Na_2_CO_3_ was added to inactivate the enzyme. The *o*-nitrophenol (*o*NP) was observed at 420 nm. One unit of enzyme activity was defined as hydrolyzing 1 μmol *o*NPG to *o*NP per minute.

#### 3.5.3. Immunofluorescence Microscopy and Flow Cytometry Analysis

To verify β-galactosidase display on the surface of *P. pastoris*, the recombinant strains were observed under an immunofluorescence microscope (Mshot, MF52-N, Guangzhou, China) and with flow cytometry (CytoFLEX, Beckman Coulter, Bria, California, USA). After 120 h of methanol-induced fermentation, 1 mL cells were collected by centrifugation at 6000 rpm for 5 min, washed three times with PBS (pH 7.4), and suspended in PBS containing 2% BSA (pH 7.4). Then, 1 μL FLAG-tagged mouse monoclonal antibody (1:100) was added to a suspension of 200 μL. After incubation for 3 h at 37 °C and shaking at 60 rpm, the cells were washed three times with PBS and suspended in PBS containing 2% BSA. Then, 1 μL FITC-labeled goat anti-mouse IgG antibody (1:500) was added to the cells and incubated for 3 h at 37 °C with shaking at 60 rpm. The cells were washed again with PBS three times and observed under an immunofluorescence microscope. Additionally, 200 μL cells were filtered through a 600 mesh sieve (aperture is 0.025 mm) to disperse the cell clusters, and the samples were measured with a flow cytometer. We collected 10,000 cells for the assay using an excitation light wavelength was 488 nm, and the channel signal was recorded [39].

#### 3.5.4. Enzymatic Properties of β-Galactosidase

One milliliter of cells was collected after 72 h of induced expression, centrifuged at 6000 rpm for 5 min to remove the supernatant, and the cells were washed three times with PBS. To investigate the effect of temperature on enzyme activity, 800 μL 20 mmol/L *o*NPG and pH 6.0 citrate buffer was added for 10 min at 30–80 °C. The β-galactosidase enzyme activity was measured at 420 nm at the end of the reaction. To investigate the effect of pH on enzyme activity, the reaction was conducted in a 50 °C water bath at pH 3.0 to 9.0. The same number of cells were taken and held at 40–70 °C and pH 4.0–7.0 for 120 min, with samples taken every 30 min to measure the enzyme activity. Samples were also taken at the optimum temperature and pH to measure the activity and half-life of the enzyme. In addition, 5 mmol/L, 25 mmol/L, and 50 mmol/L Na^+^, Ca^2+^, Mg^2+^, Fe^2+^, Fe^3+^, Mn^2+^, Co^2+^, Cu^2+^, and Zn^2+^ were added to pH 6.0 buffer and the enzyme activity of β-galactosidase was measured after 10 min of reaction at 50 °C in a water bath.

### 3.6. Free and Immobilized Fermentation for GOS Synthesis

The dry weight of the cells at the end of fermentation was measured to ensure uniformity in the experiment. For cells on immobilized carriers, 100 mL carriers and free cells at the end of fermentation was washed twice with PBS buffer, centrifuged at 6000 rpm for 5 min, dried at 105 °C, and weighed. To determine the optimum substrate concentration and reaction time for GOS production by fermentation, concentrations ranging from 300–600 g/L lactose were made in citrate buffer (pH 6.0). The substrate was catalyzed at 40 °C by strain Δ0226-Pir1p-lacA. The samples were taken every 3 h. The GOS, glucose, galactose, and lactose content were analyzed using HPLC.

In order to optimize the catalytic process and verify whether the cells could achieve growth and catalysis at the same time, citrate buffer (pH 6.0) and 20–100% BMMY (pH 6.0) were added to the buffer. After 4 batches of catalysis, the production of GOS and enzyme activity of the cells were tested. The number of reused free and immobilized cells was examined using the same catalytic system, to which the lactose substrate was re-added every 6 h to determine the lactose conversion rate. When the catalytic efficiency of the immobilized cells dropped below 70%, the immobilized carriers were placed back into a fresh medium to recover, then taken out to continue catalysis. This cycle was repeated several times until the catalytic effect was greatly reduced. The total time–space yield of free and immobilized cells was calculated as follows:① Free cells (g/L/h) = $\frac{\mathrm{Total}\;\mathrm{production}\;\mathrm{of}\;\mathrm{GOS}}{\mathrm{Total}\;\mathrm{cell}\;\mathrm{growth}\;\mathrm{time}+\mathrm{catalytic}\;\mathrm{time}}$② Immobilized cells (g/L/h) = $\frac{“\mathrm{Total}\;\mathrm{production}\;\mathrm{of}\;\mathrm{GOS}”\;}{“\mathrm{Total}\;\mathrm{cell}\;\mathrm{growth}\;\mathrm{time}+\mathrm{resuscitation}+\mathrm{catalytic}\;\mathrm{time}”\;}$

### 3.7. The Methods of Product Detection

The analysis of glucose, galactose, and GOS was performed on an Agilent 1260 Infinity II RID instrument and an Aminex HPX-87H analytical HPLC column at 60 °C with 5 mmol/L H_2_SO_4_ as the mobile phase and a flow rate of 0.4 mL/min.

## 4. Conclusions

The biofilm-based fermentation of *P. pastoris* was developed by using the advantages of biofilm formation. Knocking out the cell wall protein gene *PAS_chr1-3_0226* successfully enhanced biofilm formation, which provided excellent growth conditions for the cells to maintain high activity during prolonged fermentation, which was applicable to continuous (repeated-batch) fermentation. In addition, the optimal surface-displayed strains with anchorin were selected based on strain GS115-Δ0226, with the enzyme activity of 5125 U/g and 23 repeats of continuous catalysis batches. This could improve catalytic efficiency and repeated batches compared with the free fermentation. A co-culture system integrated catalysis and growth, so that a dynamic equilibrium in the consolidated integrative process was achieved. These results indicated that biofilm-based fermentation using surface display will be of great value for immobilized GOS production. Overall, this study provides a reference for the development of more biofilm-based surface display methods in *P. pastoris* for biochemical production processes.

## Figures and Tables

**Figure 1 ijms-24-06507-f001:**
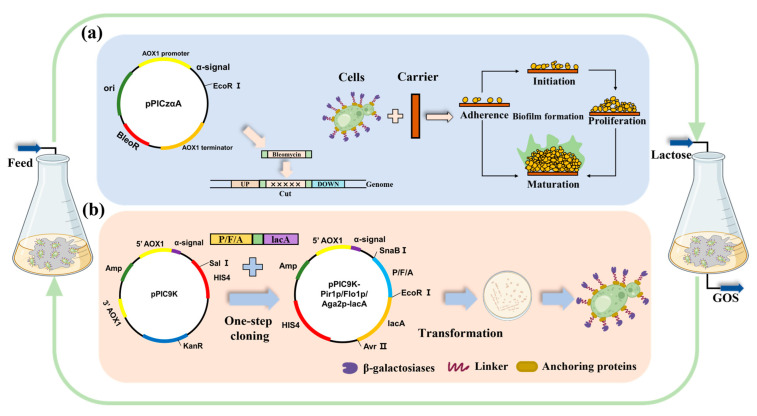
Biofilm-based biocatalysis for surface-displayed strains in dynamic growth and catalytic GOS production. (**a**) A schematic illustration of construction for the knockout of *PAS_chr1-3_0226* and biofilm formation in *P. pastoris*. (**b**) The DNA components of the surface-displayed plasmids used in this study and a schematic diagram of the yeast surface display.

**Figure 2 ijms-24-06507-f002:**
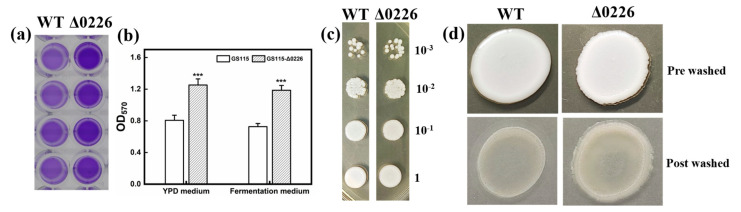
The impact of the knockout of *PAS_chr1-3_0226* gene on biofilm formation and cell adhesion. (**a**) GS115 and GS115-Δ0226 strains were cultivated in 96-well plates for 72 h and tested for adhesion ability. Free cells were removed and washed twice with PBS buffer and stained with 0.1% crystal violet. The samples were washed repeatedly with water, lysed with glacial acetic acid, and then photographed. (**b**) Adhesion expressed as the optical density at 570 nm (OD_570_) of solubilized crystal violet in acetic acid. Data are reported as the means and standard deviation of three independent experiments. The *p*-values were computed using Student’s *t*-test (*** *p* < 0.001). (**c**) Growth of GS115 and GS115-Δ0226 on YPD at an OD_600_ = 1 and used to make 10^−1^, 10^−2^, and 10^−3^ gradient dilutions in sterile water. (**d**) Standard plate-wash assay of GS115 and GS115-Δ0226.

**Figure 3 ijms-24-06507-f003:**
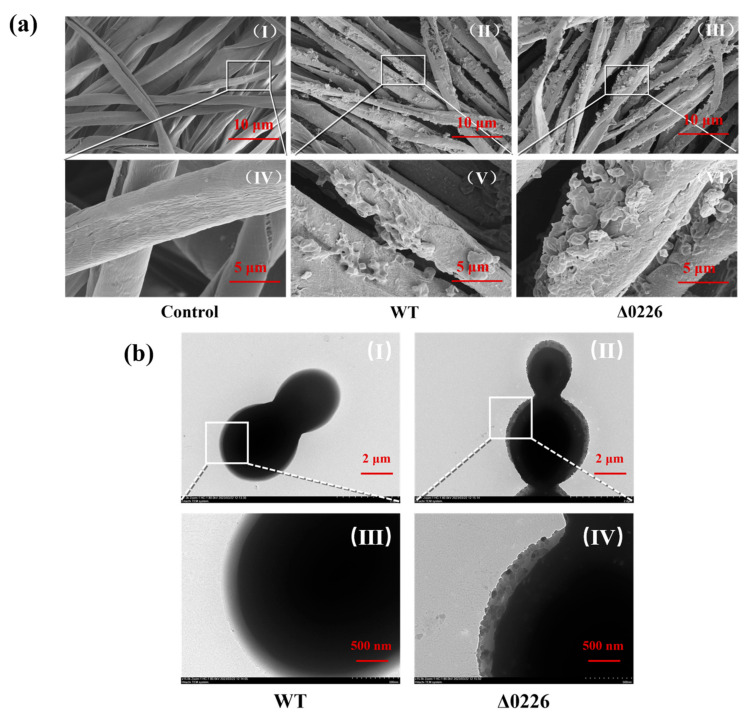
SEM (**a**) and TEM (**b**) images of cells after 48 h of fermentation. (**a**) The biofilm formation on the cotton fibers of the control, GS115 wild type, and GS115-Δ0226. The observation scale was 10 μm in I to III and 5 μm in IV to VI. The deletion of *PAS_chr1-3_0226* could enhance biofilm formation. (**b**) The microstructure of cell morphology of GS115 wild type and GS115-Δ0226. WT was normal with smooth surface, while strain Δ0226 was rough and the specific surface area was increased. The observation scale was 2 μm in I to II and 500 nm in III to IV.

**Figure 4 ijms-24-06507-f004:**
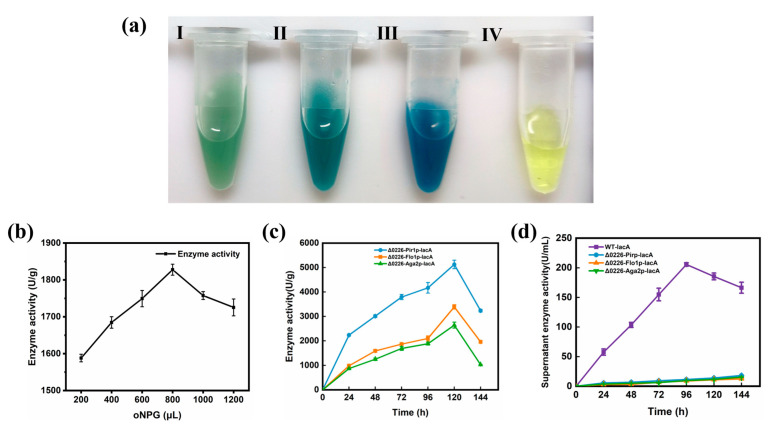
Expression and enzyme activity of surface-displayed β-galactosidase in *P. pastoris*. (**a**) β-galactosidase expression in (I) Δ0226-Aga2p-lacA, (II) Δ0226-Flo1p-lacA, (III) Δ0226-Pir1p-lacA, and (IV) the control. The darker the color of the sample, the higher the expression of β-galactosidase. (**b**) Enzyme activity was determined by adding 200–1200 μL *o*NPG to precipitate to determine the optimum substrate concentration. (**c**) The enzyme activities of precipitate by the strains Δ0226-Pir1p-lacA, Δ0226-Flo1p-lacA, and Δ0226-Aga2p-lacA. (**d**) The enzyme activity of supernatant by the free expression of WT-lacA and the three surface-displayed strains.

**Figure 5 ijms-24-06507-f005:**
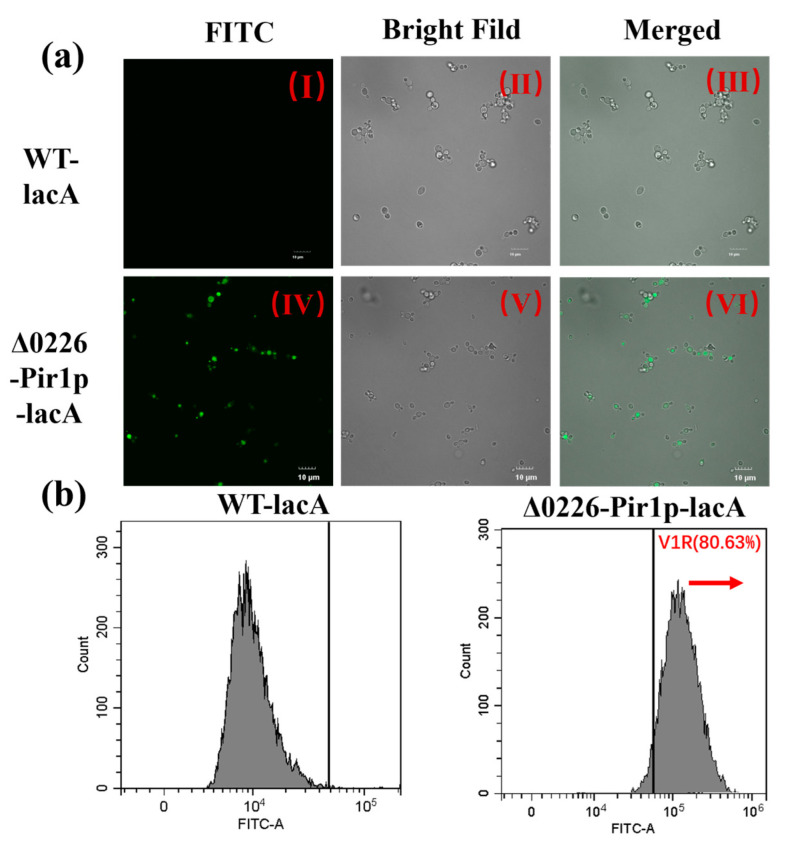
The characterization of β-galactosidase by (**a**) immunofluorescence microscopy and (**b**) flow cytometry in WT-lacA and Δ0226-Pir1p-lacA strains. (**a**) Panels Ⅰ and Ⅳ show representative immunofluorescence micrographs of the free expression WT-lacA and surface-displayed strain Δ0226-Pir1p-lacA using FITC excitation wavelengths; Ⅱ and Ⅴ show the visible light images, and Ⅲ and Ⅵ are merged composites of the visible light and FITC excitation images. All micrographs shown were taken at 1000× magnification and at the same angle. Scale bar = 10 μm. (**b**) Flow cytometry histograms reflect the fluorescence signal of the surface-displayed strain, with the *x*-axis indicating fluorescence intensity and the *y*-axis indicating the cell count. Red arrow: the peak shifted to the right indicated that the efficiency of surface display reached 80.63%.

**Figure 6 ijms-24-06507-f006:**
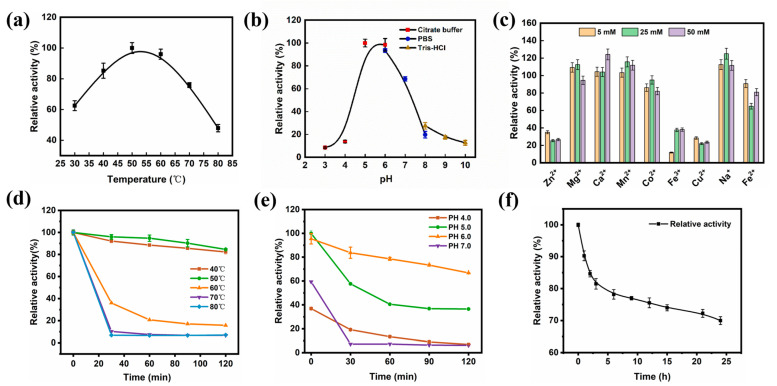
The characterization of surface-displayed β-galactosidase enzyme activity. (**a**) The influence of temperature on the relative activity of recombinant strain. (**b**) Effect of pH on the relative activity of recombinant strain. (**c**) Effect of exogenous addition by different concentrations of metal ions at 5 mmol/L, 25 mmol/L, and 50 mmol/L on the relative activity of recombinant strain, calculated assuming 100% enzyme activity without the addition of metal ions. (**d**) The surface-displayed enzyme was maintained at 40–80 °C for 120 min and the relative enzyme activity was measured every 30 min. (**e**) The surface-displayed enzyme was maintained at pH 4.0–7.0 for 120 min and the relative enzyme activity was measured every 30 min. (**f**) The surface-displayed enzyme was held at pH 6.0 and 50 °C for 24 h and sampled at intervals to measure the enzyme activity. Calculations assumed 100% activity at 0 h.

**Figure 7 ijms-24-06507-f007:**
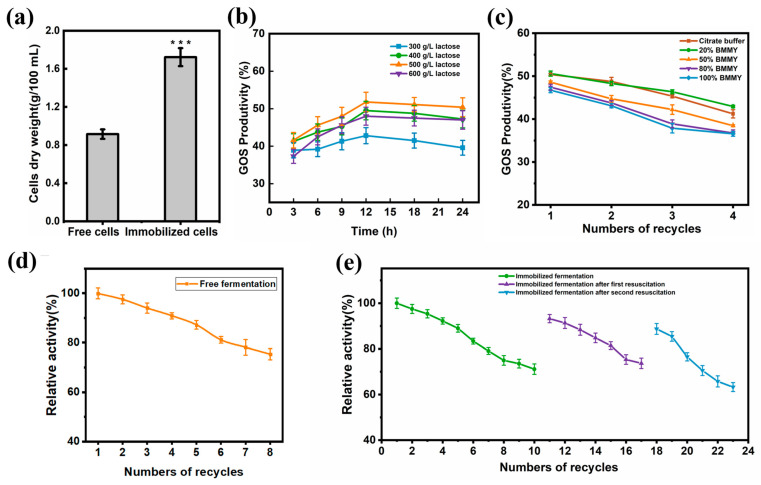
Optimization of GOS production and cell reuse times. (**a**) Comparison of immobilized and free cell dry weights. Data are reported as the means and standard deviation of three independent experiments. The *p*-values were computed using Student’s *t*-test (*** *p* < 0.001). (**b**) Effect of lactose concentration and catalytic time on GOS production by the surface-displayed strains. GOS production was increased in a lactose concentration-dependent manner, with the highest production of 251 g/L GOS at 500 g/L lactose (50.3% yield), before a sudden drop in production at concentrations above 500 g/L lactose. The optimal production time for GOS was 12 h, after which the yield stabilizes. (**c**) Influence of different volumes of BMMY medium on GOS yield in multi-batch catalysis. A total of 4 catalysis batches were conducted, with the highest catalytic efficiency starting in the third batch with 20% BMMY. Adding more than 20% BMMY negatively impacted GOS yield. (**d**) Number of reuses in the free cell catalysis. (**e**) Number of reuses in the immobilized cell catalysis. When the GOS yield drops to a certain level, the immobilized cells were resuscitated for 48 h and before catalysis was resumed.

**Table 1 ijms-24-06507-t001:** Analysis of reaction products.

	Yield of GOS (%)							Time-Space Yields of GOS (g/L/h)
	Lactose	Glucose	Galactose	Transfer Disaccharide	Trisaccharide	Tetrasaccharides	GOS	
Free surface-displayed enzyme	34.2 ± 0.2	10.6 ± 0.3	6.6 ± 0.6	15 ± 0.2	24.6 ± 0.1	9 ± 0.2	48.6 ± 0.6	0.33 ^a^
Immobilized surface-displayed enzyme	33.8 ± 0.2	9.8 ± 0.2	6 ± 0.3	18.1 ± 0.1	23.6 ± 0.2	8.6 ± 0.1	50.3 ± 0.3	0.44 ^b^

^a^: Total GOS production: 3934.5 g, catalytic volumes: 24 L, total time: 504 h. ^b^: Total GOS production: 3613.7 g, catalytic volumes: 23 L, total time: 354 h.

**Table 2 ijms-24-06507-t002:** Strains and plasmids used in this study.

Strains or Plasmids	Relevant Characteristics	Sources
Strains		
*Escherichia coli*	DH5α	Stored in our lab
GS115	*P. pastoris*	Stored in our lab
GS115-Δ0226	*P. pastoris* with the deletion of *PAS_chr1-3_0226*	This study
GS115-lacA	*lacA* comes from *Aspergillus oryzae* BK03	This study
Δ0226-Pir1p-lacA	Pir1p comes from *S. cerevisiae* S288c	This study
Δ0226-Aga2p-lacA	Aga2p comes from *S. cerevisiae* S288c	This study
Δ0226-Flo1p-lacA	Flo1p comes from *S. cerevisiae* S288c	This study
Plasmids		
pPIC9K	Resistance to Ampicillin	Stored in our lab
pPICZαA	Resistance to Bleomycin	Stored in our lab

**Table 3 ijms-24-06507-t003:** Primers used in this study and their sequences.

Primer Name	Primer Sequence	Source
Δ0226-UP-F	CCTAGTGATTCCTGTGATGTATTCACGGCTGCGCAAAACT	This work
Δ0226-UP-R	GCTATGGTGTGTGGGGGATCATTTTGATTATCTTTGTGAG	This work
Δ0226-DOWN-F	GCTCGAAGGCTTTAATTTGCGCGGTTCACATTAATTAAAG	This work
Δ0226-DOWN-R	AAATTTAAAGAGATGCGAAACTTGACAGCTTGAGCGTGAC	This work
BleoR-F	CTCACAAAGATAATCAAAATGATCCCCCACACACCATAGC	This work
BleoR-R	CTTTAATTAATGTGAACCGCGCAAATTAAAGCCTTCGAGC	This work
Pir1p-UP	AAAGAGAGGCTGAAGCTTACGTATATGCTCCAAAGGACCC	This work
Pir1p-DOWN	CCAGAACCACCACCACCGAATTCACAGTTGAGCAAATCGA	This work
lacA-UP	GCTCAACTGTGAATTCGGTGGTGGTGGTTCTGGTGGTGGTGGA TCTGGTGGTGGAGGTTCTTCTATTAAGCATAGA	This work
lacA-DOWN	TTAATTCGCGGCCGCCCTAGGTTACTTATCATCATCATCCTTGTA ATCGTAAGCACCCTTTCTT	This work
FS-UP	TACGTAGCCACAGAGGCGTGCTTACCAGCAGGCCAGAGGAAAA	This work
FS-DOWN	GAATTCAGAGCTGGTGATTTGTCCTGAAGATGATGATGAC	This work
Aga2p-UP	AAAGAGAGGCTGAAGCTTACGTACAGGAACTGACAACTAT	This work
Aga2p-DOWN	CCAGAACCACCACCACCGAATTCAAAAACATACTGTGTGT	This work
Δ0226-UP100 bp	TGAGACACATTTAACCATCGC	This work

The lacA-UP redlined section is a 45 bp linker, and the lacA-DOWN redlined section is a 24 bp FLAG tag.

## Data Availability

Not applicable.

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
