# Peer review of "Biofilm-Based Biocatalysis for Galactooligosaccharides Production by the Surface Display of β-Galactosidase in Pichia pastoris"

_ijms, 2023, doi:10.3390/ijms24076507_

Round 1

Reviewer 1 Report

plaese provide the XRD pattern for synthesized galactooligosaccharides with charactrisize them. 

authors also can provide the TEM image for morphology detection. 

these references can be cited in the manuscript

Sulfonamide-functionalized covalent organic framework (COF-SO3H): an efficient heterogeneous acidic catalyst for the one-pot preparation of polyhydroquinoline and 1,4-dihydropyridine derivatives. Farsi, R.M.K.Mohammadi,  Saghanezhad, S.J. Research on Chemical Intermediates202147(3)pp. 1161–1179

Biocatalytic strategies in the production of galacto-oligosaccharides and its global status .Parmjit S. Panesar , Rupinder Kaur , Ram S. Singh , John F. Kennedy. International Journal of Biological Macromolecules111, May 2018, Pages 667-679

 regards

Author Response

  1. please provide the XRD pattern for synthesized galactooligosaccharides with characterize them.

Fig Detection of catalytic products by HPLC

The characterization and quantification of galactooligosaccharides were operated by HPLC. The appearance time and peak area of standard substances were detected by external standard method. The methods were in Line 501 to 504. Hence, it also could characterize the synthesis of galactooligosaccharides.

  1. authors also can provide the TEM image for morphology detection.

After receiving notification of revision, TEM test of different strains was carried out. We had added more description and analysis in Line 131 to 147. SEM and TEM were merged in Fig. 3.

The revised part was as follows:

In order to investigate the microstructure changes of cell morphology, TEM test was carried out. Generally speaking, the beta-1,3-glucanosyltransferase encoded by PAS_chr1-3_0226 was required for cell wall assembly in Pichia pastoris. The results of TEM showed that the morphology of WT was normal with smooth surface (Figure 3b I, III). Meanwhile, the substances similar to biofilm matrix was observed on the sur-face of strain Δ0226 (Figure 3b II, IV). These results were consistent with the conclu-sion that the deletion of PAS_chr1-3_0226 could enhance biofilm formation in Figure 2. The surface of strain Δ0226 was rough and the specific surface area was increased, which could be beneficial to form more anchorin on the surface of the cell wall in the following surface display.

Figure 3. SEM (a) and TEM (b) images of cells after 48 h of fermentation. (a) The biofilm formation on the cotton fibers of the control, GS115 wild type and GS115-Δ0226. The observation scale was 10 μm in I to III and 5 μm in IV to VI. The deletion of PAS_chr1-3_0226 could enhance biofilm for-mation. (b) The microstructure of cell morphology of GS115 wild type and GS115-Δ0226. WT was normal with smooth surface, while strain Δ0226 was rough and the specific surface area was in-creased. The observation scale was 2 μm in I to II and 500 nm in III to IV.

  1. these references can be cited in the manuscript

Sulfonamide-functionalized covalent organic framework (COF-SO3H): an efficient heterogeneous acidic catalyst for the one-pot preparation of polyhydroquinoline and 1,4-dihydropyridine derivatives. Farsi, R., M.K.Mohammadi,  Saghanezhad, S.J. Research on Chemical Intermediates, 2021, 47(3), pp. 1161–1179

Biocatalytic strategies in the production of galacto-oligosaccharides and its global status .Parmjit S. Panesar , Rupinder Kaur , Ram S. Singh , John F. Kennedy. International Journal of Biological Macromolecules. 111, May 2018, Pages 667-679

The first reference was about XRD test, while this manuscript was not related to XRD. The detection method was HPLC instead of XRD. The second reference was cited in [3] in Line 38.

Reviewer 2 Report

The manuscript by Chen et al  reports the combination of the surface displayed system and biofilm immobilized fermantation using Pichia pastoris cells to catalyse the formation of galactooligosaccharides using β-galactosidase activity. Biofilm-based biocatalysis showed highly efficient productivity of galactooligosaccharides, the shortened culture time and persistent cell activity. The work described is very well performed and the experimental part is clear and concise. The quality of the results presented is high, and the concept of the paper is interesting.

Minor suggestion:

Avoid expressing conversions/yields with decimals e.g. 50.3 % since it is not accurate. The intrinsic error of techniques such as GC or HPLC is 1 %, therefore, present values with decimals do not have scientific meaning (instead 50 %).

Author Response

The manuscript by Chen et al. reports the combination of the surface displayed system and biofilm immobilized fermantation using Pichia pastoris cells to catalyse the formation of galactooligosaccharides using β-galactosidase activity. Biofilm-based biocatalysis showed highly efficient productivity of galactooligosaccharides, the shortened culture time and persistent cell activity. The work described is very well performed and the experimental part is clear and concise. The quality of the results presented is high, and the concept of the paper is interesting.

  1. Avoid expressing conversions/yields with decimals e.g. 50.3 % since it is not accurate. The intrinsic error of techniques such as GC or HPLC is 1 %, therefore, present values with decimals do not have scientific meaning (instead 50 %).

We revised Table 1 and added the error to show the scientific meaning of conversions/yields.

Table 1 Analysis of reaction products

Yield of GOS (%)

Time-space yields of GOS (g/L/h)

Lactose

Glucose

Galactose

Transfer disaccharide

Trisaccharide

Tetrasaccharides

GOS

Free Surface-displayed enzyme

34.2 ± 0.2

10.6 ± 0.3

6.6 ± 0.6

15 ± 0.2

24.6 ± 0.1

9 ± 0.2

48 ± 0.6

0.33a

Immobilized surface-displayed enzyme

33.8 ± 0.2

9.8 ± 0.2

6 ± 0.3

18.1 ± 0.1

23.6 ± 0.2

8.6 ± 0.1

50 ± 0.3

0.44b

a: Total GOS production: 3934.5 g, Catalytic volumes :24 L, Total time: 504 h.

b: Total GOS production: 3613.7 g, Catalytic volumes :23 L, Total time: 354 h
